# Is Piperacillin-Tazobactam an Appropriate Empirical Agent for Hospital-Acquired Sepsis and Community-Acquired Septic Shock of Unknown Origin in Australia?

**DOI:** 10.3390/healthcare10050851

**Published:** 2022-05-05

**Authors:** Alice Gage-Brown, Catherine George, Jenna Maleki, Kasha P. Singh, Stephen Muhi

**Affiliations:** 1Victorian Infectious Diseases Service (VIDS), The Royal Melbourne Hospital, Parkville, VIC 3050, Australia; alice.gagebrown@mh.org.au (A.G.-B.); kasha.singh@mh.org.au (K.P.S.); 2Pharmacy Department, The Royal Melbourne Hospital, Parkville, VIC 3050, Australia; catherine.george@mh.org.au (C.G.); jenna.maleki@mh.org.au (J.M.)

**Keywords:** piperacillin-tazobactam, sepsis, empirical

## Abstract

Early appropriate empirical antibiotics are critical for reducing mortality in sepsis. For hospital-acquired sepsis of unknown origin in Australia, piperacillin-tazobactam (TZP) is recommended as an empirical therapy. Anecdotally, some institutions also use TZP for community-acquired septic shock. This narrative review aimed to scrutinise the appropriateness of TZP as an empirical agent for undifferentiated hospital-acquired sepsis and community-acquired septic shock. An online database (Medline) was searched for relevant studies in adults published in the last 10 years. Studies were included if they addressed separately reported clinical outcomes related to a relevant aspect of TZP therapy in sepsis. Of 290 search results, no studies directly addressed the study aim. This review therefore explores several themes that emerged from the contemporary literature, all of which must be considered to fully interrogate the appropriateness of TZP use in this context. This review reveals the paucity and low quality of evidence available for TZP use in sepsis of unclear origin, while demonstrating the urgent need and equipoise for an Australian audit of TZP use in patients with sepsis of unknown origin.

## 1. Introduction

Sepsis is a serious infectious syndrome caused by myriad pathogens and is a significant contributor to mortality, morbidity, and healthcare costs in Australia and worldwide. Sepsis is defined by The Third International Consensus Definitions for Sepsis and Septic Shock as ‘life-threatening organ dysfunction caused by a dysregulated host response to infection’ [1]. In clinical settings, sepsis can be rapidly identified in patients with suspected infection plus at least two criteria of the quick Sequential Organ Failure Assessment (qSOFA) tool: respiratory rate ≥22/min, altered mentation, or systolic blood pressure ≤100 mmHg [1]. Septic shock is diagnosed when vasopressors are required to maintain mean arterial pressure ≥65 mmHg and serum lactate is ≥2 mmol/L despite adequate volume resuscitation [1]. Every year, 11 million or almost one-fifth of all deaths worldwide are associated with sepsis [2]. Of those who survive, many live with permanent disability. In 2017, there were 104,912 hospitalisations with sepsis Australia-wide, of which 11% resulted in death in hospital, with an estimated cost per episode of AUD 39,300 [3,4]. In 2020, 69% of sepsis cases in Australia were due to gram-negative pathogens, with the most common being *Escherichia coli* [5]. Gram-positive pathogens, most commonly *Staphylococcus aureus*, accounted for the remaining 31% of sepsis cases [5].

Early and adequate empirical antibiotics are a key predictor of outcome in sepsis [6]. The choice of an appropriate empirical agent is based on a range of factors including hospital or community acquisition, septic focus, illness severity, allergy status, local antimicrobial resistance patterns, patient risk factors for antimicrobial resistance, and patient co-morbidities such as impaired renal function. If the source of infection is not immediately apparent, termed ‘sepsis of unknown origin’, empirical antibiotics must cover a wide range of possible pathogens. For hospital-acquired sepsis of unknown origin, the Australian Therapeutic Guidelines: Antibiotic Version 16 recommend empirical intravenous (IV) piperacillin-tazobactam (TZP) as first-line therapy for patients with no penicillin hypersensitivity [7]. Anecdotally, some institutions also prescribe TZP as empirical therapy for community-acquired septic shock, although this is not recommended by the Australian Therapeutic Guidelines.

TZP is a broad-spectrum IV antibiotic composed of piperacillin, a ureidopenicillin with extended spectrum activity, and tazobactam, a beta-lactamase inhibitor [8]. Piperacillin exerts bactericidal action by inhibiting penicillin-binding proteins, thereby disrupting bacterial cell wall cross-linkage. Tazobactam prevents hydrolysis of piperacillin by forming a stable acyl-enzyme complex with bacterial beta-lactamases [8]. In 2017, TZP was the seventh most used antibiotic in Australia, making up 4.6% of total antibiotic usage. This figure is probably an underestimate, as TZP use was restricted in 2017 owing to a national shortage [9]. As a first-line empirical agent, TZP has established activity against many gram-positive, gram-negative, and anaerobic bacteria [8]. To address gaps in bacterial coverage, the Australian Therapeutic Guidelines have recommended adding IV vancomycin to cover methicillin-resistant *S. aureus* in septic shock or line-related sepsis and substituting TZP for IV meropenem when there are risk factors for multi-drug-resistant (MDR) gram-negative bacteria [7].

Emerging resistance mechanisms in gram-negative bacteria raise concerns regarding the continued appropriateness of TZP as a first-line broad spectrum antibiotic. In Australia in 2020, 14.7% of *E. coli* and 10.0% of *Klebsiella pneumoniae* bloodstream isolates had extended-spectrum beta-lactamase (ESBL) phenotype and 8.3% of bloodstream isolates were ‘ESCAPPM’ AmpC producers (*Enterobacter, Serratia, Citrobacter freundii, Acetinobacter/Aeromonas, Providencia, Proteus vulgaris*, and *Morganella* species) [5]. In 2020, 11.3% of *Pseudomonas aeruginosa* isolates tested in Australia were resistant to antibiotics of at least two classes [5]. ESBLs derive from mutations in chromosomal or plasmid-borne beta-lactamase genes and confer resistance to third generation cephalosporins and aztreonam [10]. Although carbapenems are the treatment of choice for ESBL-producing Enterobacteriaceae (ESBL-PE), the alarming emergence of carbapenem resistance has led to renewed consideration of carbapenem-sparing agents such as TZP [7,11]. AmpC beta-lactamases confer resistance to many antibiotics including TZP and are impervious to beta-lactamase inhibitors, including tazobactam [12]. AmpC beta-lactamases are chromosomally encoded in ESCAPPM bacteria [12]. The three primary resistance mechanisms for AmpC beta-lactamases are inducible chromosomal expression, stably derepressed chromosomal expression and plasmid-mediated resistance [12]. Even if initially beta-lactam susceptible, exposure to beta-lactams may temporarily lead to resistance in ESCAPPM species via inducible AmpC beta-lactamase production [12]. TZP is thought to be a weak inducer, suggesting it may be an appropriate agent for ESCAPPM and other organisms with inducible AmpC beta-lactamase [12].

Important considerations in evaluating TZP appropriateness are toxicity and dosing. Nephrotoxicity and neurotoxicity are potential adverse events reported in the literature, but studies in patients with defined sepsis are limited [13,14]. Appropriate dosing regimens aim to optimise clinical outcomes in patients with adverse risk factors including immunosuppression, infections with a high bacterial inoculum (e.g., deep-seated abscess or collection) or with a high minimum inhibitory concentration (MIC) to TZP. Dose optimization may also be appropriate for patients with augmented renal clearance, as well as critically ill and obese patients. In Australia, TZP is conventionally infused over 30 min and serum drug measurements are not routinely recommended [7].

The aim of this review is to address the appropriateness of TZP as an empirical agent for hospital-acquired sepsis and community-acquired septic shock of unknown origin in Australia. Because of a gap in the literature for efficacy of empirical TZP in sepsis of unknown origin in Australia, this review will focus more broadly on clinical outcomes of empirical or definitive TZP for sepsis of any source in any country. Moreover, this review will not separate clinical outcomes related to hospital versus community-acquired sepsis and sepsis versus septic shock, as these terms are often variably and poorly defined in the literature and are rarely analysed as separate entities. Therefore, this review summarises the contemporary literature surrounding clinical outcomes related to empirical or definitive TZP efficacy, toxicity, and dosing considerations in sepsis of any source. Subsequently, these findings are expected to establish the basis of a study addressing the original research question, relating to hospital-acquired sepsis and community-acquired septic shock of unknown origin in the Australian setting.

## 2. Materials and Methods

The online database Medline (Ovid) was searched on 13 August 2021 using MeSH terms and key words for ‘sepsis’ and ‘TZP’ and publication date from 2011 to 2021. Search history is listed in full in Appendix B.

Studies were excluded on title and abstract and full text screening for the following reasons:Written in a language other than English;Meta-analysis or study protocol;Full text unavailable online;The study exclusively concerned a paediatric population;Outcomes relating to either TZP therapy or sepsis were not reported separately from other outcomes;Study participants did not have known or suspected sepsis, as defined by positive blood culture, clinical diagnosis of sepsis, bacteraemia, or bloodstream infection;The study did not address clinical outcomes related to a relevant aspect of TZP therapy; efficacy, resistance, toxicity, or dosing regimens.

Studies were included in the review if a major aim of the study was to investigate clinical outcomes of TZP in sepsis in adults. In order to specifically address the rapidly emerging contemporary evidence on the clinical efficacy of TZP in ESBL and AmpC beta lactamase-producing gram-negative pathogens, an additional search was performed using the key words ‘piperacillin-tazobactam’ and ‘ESBL’ or ‘AmpC’ on 26 April 2022.

The latest Antimicrobial Use and Resistance in Australia (AURA) and sepsis outcome program reports were retrieved from safetyandquality.gov.au (accessed on 14 August 2021) and included to assess the relevance of international studies in the Australian context. Additional studies referenced only in the introduction were retrieved from the reference lists of included articles.

## 3. Results

No studies identified from the database search directly addressed sepsis of unknown origin or the appropriateness of TZP monotherapy in sepsis of any source. One study addressed the appropriateness of TZP/vancomycin combination therapy in sepsis [15] but was excluded as the appropriateness of TZP alone was not reported separately. The following studies were included in this review: 24 addressing clinical efficacy of TZP (Table 1), 4 reporting toxicity (Table 2), and 5 exploring pharmacokinetic and dosing considerations (Table 3). Additional case reports are included in Appendix A.

## 4. Discussion

This review addresses the appropriateness of TZP as an empirical agent for sepsis of unknown origin from three angles; clinical efficacy against known pathogens, reported toxicities in septic patients, and optimised dosing.

### 4.1. Efficacy against ESBL-Producing Enterobacteriaceae

In 2019, more than 1 in 10 Australian *E. coli* and *K. pneumoniae* blood isolates harboured ESBLs [5], but evidence is mixed regarding the utility of TZP as a carbapenem-sparing agent for these pathogens. Following the results of a recent randomised controlled trial (RCT), the Australian Therapeutic Guidelines recommended substituting TZP for meropenem where risk factors for ESBL producers are present [7]. However, several retrospective studies report conflicting results, with most supporting TZP as a useful alternative to carbapenems.

Harris et al. conducted a noninferiority RCT in nine countries, including Australia, that concluded that definitive TZP, compared with meropenem, was associated with increased 30-day mortality [32]. However, this conclusion may not apply to empirical TZP and septic shock, as only definitive therapy was investigated, and most study participants had low illness severity. Furthermore, the study was limited by lack of blinding and was underpowered to analyse subgroups where definitive TZP may have been noninferior to meropenem. A post-hoc analysis found that Harris et al. may have overestimated mortality in the TZP group by inclusion of isolates susceptible to TZP by automated testing methods that were found to be TZP non-susceptible post hoc by broth microdilution susceptibility testing [32,33]. When TZP non-susceptible strains were excluded, there was no longer any significant difference in mortality between TZP and meropenem treatment arms [33]. Isolates co-harbouring ESBL and narrow spectrum OXA-type beta lactamases were associated with increased TZP MIC and elevated mortality [33]. Therefore, rapid identification of beta lactamase type alongside susceptibility testing may guide the selection of appropriate definitive antibiotics. However, a larger study is needed to explore the association between mortality and other beta lactamase types in isolates with ESBL phenotype. In the absence of further data, meropenem should continue to be the first-line therapy for patients with sepsis with confirmed or suspected ESBL *Enterobacteriaceae.*

Out of eight retrospective observational studies directly comparing TZP and carbapenems in ESBL-PE sepsis, only two found an association between empiric TZP use and mortality [26,28]. One study in patients with haematological malignancy found that although empiric TZP was not linked to increased mortality, persistent bacteraemia and prolonged fever were more common than in the carbapenem group [29]. The two largest retrospective studies, with over 200 participants each, reported conflicting results. Tamma et al. concluded that the adjusted 14-day mortality risk was 1.92 times higher for patients receiving empiric TZP vs. carbapenem, while Ko et al. concluded no difference in 30-day mortality [28,35]. However, it is difficult to compare the two studies, as they differ in several important aspects: geographical location, sample size of the TZP group, and infection focus. Moreover, unlike Ko et al., Tamma et al. was a single-centre study, and did not exclude patients receiving the study antibiotic for less than 48 h, possibly obscuring the effect of the empirical therapy. Note also that the lower bound of the 95% confidence interval for the finding of 14-day mortality risk was 1.07, which although statistically significant, may be clinically irrelevant [28].

In two retrospective studies, lower TZP MIC was associated with reduced mortality in TZP-treated ESBL-PE sepsis than higher MIC, even within the Clinical and Laboratory Standards Institute (CLSI) susceptible range of ≤16 mg/L [30,37]. Most dramatically, Tsai et al. concluded that 30-day mortality was significantly lower for highly susceptible isolates with TZP MIC ≤ 0.5 mg/L compared to still susceptible isolates with MIC ≥ 1 mg/L, although these findings only apply to ESBL-producing *Proteus mirabilis* [37]. This supports the assertion that TZP should be used cautiously in suspected ESBL-PE sepsis. In contrast, another retrospective study observed no difference in 30-day mortality between patients with low (≤4 mg/L) and higher (8–16 mg/L) MIC in both ESBL and non-ESBL-PE sepsis [22]. However, all three studies had small overall or subgroup sample sizes, limiting the ability to adjust for confounding variables and perform meaningful MIC subgroup analyses.

Apart from the general limitations of retrospective studies discussed later, most studies discussed so far have only included TZP-susceptible infections. Therefore, the available evidence should be applied cautiously to routine clinical practice, where empirical therapy is implemented without knowing the susceptibility profile of the causative organism.

### 4.2. Efficacy against AmpC Beta-Lactamase Producers

A pilot RCT comparing meropenem and TZP for definitive treatment of bloodstream infections caused by ESCAPPM species known to possess inducible chromosomal AmpC beta-lactamases demonstrated no difference in clinical outcomes [21]. Five AmpC genotypes were identified with CMY being the most prevalent, although the degree of AmpC induction was not measured, and the sample size was too small for genotype subgroup analysis of TZP efficacy [21]. Four retrospective observational studies have found TZP to be generally as effective as cefepime or carbapenems for treating ESCAPPM bloodstream infections [16,17,18,19]. One retrospective cohort study did observe lower rates of early treatment response for patients treated with empiric TZP compared with carbapenems, but the daily dose of TZP given was relatively low (13.5 g) [20]. In all five retrospective studies, no distinction was made between chromosomal inducible, derepressed and plasmid mediated AmpC production. Notably, only one study performed AmpC genotyping and found that AmpC was detected in >95% of *Enterobacter* and *Serratia* species but in only 19% of *Citrobacter* species [17]. The remaining three studies either performed no AmpC screening or low specificity phenotypic screening, with cefoxitin-resistant isolates assumed to be AmpC producers [16,18,19]. However, McKamey et al. did find lower rates of clinical and microbiological resolution with TZP than cefepime only for isolates with both cefoxitin and third generation cephalosporin resistance, which the authors conjecture may indicate derepressed or plasmid-mediated AmpC production [18]. Given the heterogeneity in AmpC expression among ESCAPPM species, the relationship between TZP efficacy and AmpC type and level of expression is unclear and requires further research.

### 4.3. Efficacy against Pseudomonas aeruginosa

TZP appears to be a generally effective agent for sepsis caused by TZP-susceptible *P. aeruginosa*, although this is based on low-quality evidence. Only 5.5% of *P. aeruginosa* blood isolates were TZP-resistant in Australia in 2020 [5], but there are concerns that infections with MICs below the CLSI breakpoint may have poorer outcomes [50]. Reassuringly, in a case–control study of patients receiving empirical TZP, there was no difference in 30-day mortality between MICs in the CLSI low (≤16 mg/L) and intermediate (32–64 mg/L) ranges [23]. Moreover, in *P. aeruginosa* sepsis susceptible to TZP by disc testing, there was no significant difference in 30-day mortality or clinical/microbiological response between definitive TZP and other anti-pseudomonal agents, including carbapenems and ceftazidime [24]. As with many pathogens, TZP-resistance may develop within a septic episode, rendering initially successful empirical therapy ineffective. In one case report of *P. aeruginosa* sepsis associated with liver abscesses, after initial resolution of fever with empirical TZP, the patient developed a new liver abscess harbouring TZP-resistant *P. aeruginosa* [51].

### 4.4. Efficacy against Less Common Pathogens

Several case reports suggest that TZP is an appropriate antibiotic for less common pathogens involved in anaerobic and aerobic gram-negative sepsis, although this finding must be balanced against the known limitations of case reports including publication bias [52,53,54,55,56]. As an exception, TZP should be used cautiously if sepsis due to the anaerobic gram-positive bacillus *Eggerthella lenta* is suspected. A retrospective cohort study of *E. lenta* sepsis found high MICs to TZP and an independent association with 30-day mortality and Intensive Care Unit (ICU) stay in 43 patients treated with empirical TZP [38]. However, these findings are confounded by 40% of the bloodstream infections being polymicrobial. Attempts to adjust for this and other confounders by logistic regression were restricted by the small sample size. Additional case reports of less common pathogens identified in this study are available in Appendix A.

### 4.5. TZP-Related Toxicity in Sepsis

In judging the appropriateness of any empirical antibiotic, it is important to balance toxicity against benefit. There are few published reports of TZP-related toxicity in septic patients when used as monotherapy, although this review only included reports in patients with clinically or microbiologically diagnosed sepsis. TZP appears appropriate for patients at risk of renal impairment, as a large retrospective cohort study of patients with gram-negative bacteraemia found no association of TZP receipt or duration of therapy with nephrotoxicity [43]. Patients with baseline serum creatinine of >310 µmol/L were excluded, limiting the application of these findings in patients with severe pre-existing chronic kidney disease.

Neurotoxicity at high concentrations of penicillins, including piperacillin, is a known adverse effect and has been described in case reports [13], but the evidence for neurotoxic effects in patients with known sepsis is very limited. In a single-centre retrospective study of ICU patients with severe sepsis, increasing TZP trough concentrations were weakly correlated with worsening neurological status based on changes in the Glasgow Coma Scale [42]. As there are many potential causes of worsening neurological status in critically unwell patients, better controlled studies are needed to explore this correlation, preferably using more nuanced neurological examination to measure neurotoxicity.

TZP-induced thrombocytopaenia was reported in two case studies [40,41] and is an uncommon differential diagnosis to be considered in a septic patient who develops thrombocytopaenia following initiation of empirical TZP.

### 4.6. Therapeutic Drug Monitoring (TDM) and Dosing Considerations for TZP in Septic Patients

TZP may only be an appropriate empirical antibiotic for sepsis of unknown origin if dosed to maintain optimal therapeutic concentrations for sepsis due to organisms with high MIC. In *P. aeruginosa* bacteraemia, maintaining ≥60% of the dose interval above the MIC is a significant predictor of in-hospital survival [46]. Despite concerns that augmented renal clearance in sepsis would reduce the efficacy of TZP as a renally cleared drug, a prospective observational study found no difference in rate of therapeutic failure for patients with creatinine clearance of >170 mL/min [45]. However, in both these studies, steady state TZP concentrations were derived from simulated pharmacokinetic models, either based on patient characteristics such as renal function [46] or serum samples with high intra- and inter-patient variability [45]. It is therefore unclear how dose adjustments could be made based on these studies, and TDM may be required to tailor the most effective dose for each septic patient.

Extended infusion regimens, where each dose is generally infused over 4 hours instead of 30 min, may improve TZP serum concentrations without the need for individualised TDM. An RCT in a Hong Kong ICU demonstrated significantly reduced 14-day mortality in bacteraemic patients receiving extended vs. standard intermittent TZP infusion [48]. Conversely, two larger retrospective cohort studies found no overall difference in mortality rates between extended and standard infusion regimens [47,49]. Interestingly, however, Cutro et al. found that mortality rates were lower with extended infusions for patients with urinary or intra-abdominal septic foci [47]. This suggests that when septic focus is unknown extended infusion regimens may be preferred. Note that comparisons are hindered by variation in TZP doses between studies, though the doses used are generally similar to the Australian Therapeutic Guideline recommendation of 4 + 0.5 g 6-hourly [7].

### 4.7. Limitations of This Review

This review demonstrates the lack of studies directly addressing the research question. Therefore, the scope of the review was broadened to identify general themes that might be used to help frame the original research question. Nonetheless, some caveats apply to the proposed future use of these review findings in addressing the research question. Sepsis as an inclusion criterion was defined broadly as a clinical diagnosis of known or suspected sepsis, bacteraemia, or bloodstream infection or positive blood culture. Although a broad definition was needed to encompass all relevant research, patients receiving empirical TZP for sepsis of unknown origin will fulfil the clinical but not necessarily the microbiological definitions of sepsis used. Confining search results to patients with known infectious syndromes also limits knowledge of the effects of empirical TZP, particularly toxicity, on patients who have non-infectious causes for a sepsis-like presentation.

Allowing for the lack of studies concerning sepsis of unknown origin, this review focused more broadly on sepsis of any source. A future review could be used to analyse only studies with a septic focus classified as ‘unknown’ or ‘primary’. However, many studies do not stratify the analyses of TZP efficacy by infection focus, and typically, the sample sizes of patients with an unknown infection focus are small. Furthermore, such a limited analysis does not allow for recognition that a septic focus is often identified after commencement of empirical antibiotics.

This review only included studies with a major focus on clinical outcomes related to TZP therapy and may thus exclude studies with a non-clinical focus or where clinical outcomes are a minor aspect but still contribute valuable information, for example, in vitro susceptibility data for important sepsis-causing pathogens and pharmacokinetics in certain patient populations. As this review included studies investigating either empirical or definitive TZP in sepsis, the interpretation of its findings is limited when TZP is used in the critical early period before the causative organism and its antibiotic susceptibilities are known.

Apart from four RCTs and one prospective observational study, all studies discussed in this review are retrospective and observational. Unlike RCTs, retrospective observational studies are only designed to determine association, not causation, and are limited by the presence of uncontrolled confounding variables that could influence clinical outcome independently of variables of interest such as TZP treatment. Most retrospective studies reviewed here attempted to adjust for known confounders with multivariate analysis and case–control matching. However, both these techniques do not adjust for undocumented confounders and multivariate analysis has limited power with small sample sizes. Moreover, retrospective studies are more vulnerable to some biases than RCTs, such as selection, performance, and measurement biases.

The contemporary literature centres on the role of TZP in MDR sepsis compared to broader spectrum agents such as carbapenems. This focus not only biases studies towards TZP therapeutic failure, but it also fails to recognise the potential misuse of TZP in non-MDR sepsis, where, in Australia, a narrower spectrum of antimicrobials may be more appropriate. Further research is required to determine the appropriateness of TZP compared to narrower spectrum options in Australia, where sepsis of unknown origin may be caused by non-MDR organisms.

## 5. Conclusions

In conclusion, the appropriateness of TZP as an empirical agent for sepsis of unknown origin in Australia is unclear. Further research is required, particularly where antimicrobial stewardship efforts could substitute TZP for narrower spectrum agents. Despite some limitations, the highest quality evidence rejects TZP as a carbapenem-sparing agent in gram-negative ESBL producers. Limited evidence suggests that TZP may be an acceptable treatment for AmpC producers, but the impact of AmpC expression levels and genotype on TZP appropriateness is so far unknown and requires further investigation. Only low-quality studies have investigated the appropriateness of TZP in susceptible *P. aeruginosa* and anaerobic sepsis. High TZP serum concentrations have been weakly linked to neurotoxicity in ICU patients, but better controlled studies are needed to determine the clinical impact of this link. So far, there are insufficient data to support the use of extended infusion regimens of TZP in sepsis, as is reflected in the Australian guidelines. Due to the difficulties of conducting large RCTs of empirical antibiotics, the majority of studies included in this review were retrospective with small sample sizes. Although the emerging evidence is compelling, this review highlights an urgent need to prospectively evaluate the appropriateness of antibiotic interventions in the Australian setting; until then, there remains insufficient equipoise to recommend a change to current clinical practice.

## Figures and Tables

**Table 1 healthcare-10-00851-t001:** Studies addressing the clinical efficacy of TZP as monotherapy in sepsis.

Reference	Time Period	Setting	Study Design	Sample Size	Causative Pathogen	Key Findings	Limitations
Gram-negative aerobes							
Tan 2020 [16]	2010–2016	Singapore (2 centres)	Retrospective cohort study	241 (69 TZP, 69 carbapenems, 103 other antibiotics)	AmpC beta-lactamase producing Enterobacterales	No differences in 30-day mortality.	Retrospective; No screening for AmpC beta-lactamases; Only isolates susceptible to treating antibiotic included; Dosing regimens not reported
Cheng 2017 [17]	2009–2015	USA	Retrospective cohort study	165 (88 TZP, 77 cefepime/meropenem)	AmpC beta-lactamase producing Enterobacteriaceae	No difference in 7- or 30-day mortality, persistent bacteraemia, or treatment failure rate between TZP and cefepime/meropenem groups.	Retrospective; Single-centre; Small sample size post-propensity matching
McKamey 2018 [18]	2010–2015	USA	Retrospective cohort study	132 (108 cefepime, 24 TZP)	AmpC beta-lactamase-producing Enterobacteriaceae	No difference in clinical and microbiological resolution rates for TZP. Lower rate of clinical cure when isolates with baseline third generation cephalosporin and cefoxitin resistance treated with TZP.	Retrospective; Single-centre; Small sample size; Cefoxitin resistance used as a proxy for AmpC beta-lactamase production; Most had catheter-related infection; Unspecified whether empirical or definitive; Only TZP-susceptible isolates included
Drozdinsky 2021 [19]	2010–2017	Israel (2 centres)	Retrospective cohort study	277 (39 TZP, 73 third-generation cephalosporins, 61 carbapenems, 104 quinolones)	*Enterobacter* spp.	No difference in 30-day all-cause mortality between TZP and other antibiotics.	Retrospective; Small sample size; No differentiation between empirical vs. definitive therapy; No screening for AmpC beta-lactamases
Herrmann 2021 [20]	2011–2019	Germany	Retrospective cohort study	295 (81 TZP, 82 carbapenems, 132 other antibiotics)	AmpC beta-lactamase producing Enterobacterales	Treatment response within first 72 h lower for empirical TZP vs. carbapenem. Empiric TZP independently associated with treatment failure.	Retrospective; Single Centre; No screening for AmpC beta-lactamase production; Mostly respiratory infection source; Low dose of TZP used
Stewart 2021 [21]	2015–2019	Australia, Singapore, Turkey	Pilot Randomised Controlled Trial (RCT)	72 definitive therapies (38 TZP, 34 meropenem)	AmpC beta-lactamase producing Enterobacterales	No difference in 30-day all-cause mortality. Higher microbiological failure for TZP, likely confounded by inadequate source control.	Small sample size; Effect of empirical therapy unaccounted; Not all isolates screened for AmpC beta-lactamase production
Delgado-Valverde 2016 [22]	2011–2013	Spain	Prospective cohort study	275 (248 low MIC, 27 borderline MIC)	Enterobacteriaceae (ESBL and non-ESBL producers)	No difference in 30-day all-cause mortality or clinical failure for low vs. borderline MIC infections treated with empirical TZP. No increased risk in ESBL-producers.	Observational; Small sample size in borderline MIC group; Variable TZP dosing; Mostly biliary infections
Gentry 2017 [23]	2007–2013	USA	Retrospective case control study	159 (53 high MIC, 106 low MIC)	*P. aeruginosa*	No difference in 30-day all-cause mortality in low vs. intermediate MIC treated with empirical TZP.	Retrospective; Single-centre; MIC measurements not independently validated; Variable TZP dosing
Tan 2014 [24]	2007–2008	Singapore	Retrospective observational study	91 (77 monotherapy—17 TZP, 42 ceftazidime, 10 carbapenems, 8 other anti-pseudomonal)	*P. aeruginosa*	No difference in 30-day mortality, microbiological clearance, clinical response, or hospital length of stay.	Retrospective; Single-centre; Small sample sizes; Only TZP-susceptible isolates included (by disc testing only); Low illness severity
ESBL Producers							
Namikawa 2019 [25]	2011–2017	Japan	Retrospective observational study	65 empirical therapies (9 TZP, 23 carbapenem, 33 other non-carbapenem)	ESBL-PE	TZP not associated with in-hospital mortality. Trend to higher MICs in non-survivors.	Retrospective; Single-centre; Small sample size
Ofer-Friedman 2015 [26]	2010–2012	Israel, USA (multicentre)	Retrospective cohort study	79 (69 carbapenem, 10 TZP)	ESBL-PE	TZP associated with increased 90-day but not 30-day mortality.	Retrospective; Small sample size; Only TZP-susceptible infections included; Non-urinary source only
Sugimoto 2017 [27]	2009–2013	Japan (multicentre)	Retrospective observational study	35 (25 low MIC, 10 high MIC)	ESBL-PE	Better outcomes for low vs. high MIC infections treated with TZP.	Retrospective; Small sample size; Not specified whether empirical or definitive TZP
Tamma 2015 [28]	2007–2014	USA	Retrospective cohort study	213 (103 TZP, 110 carbapenem)	ESBL-PE	Higher adjusted risk of 14-day mortality for TZP vs. carbapenem.	Retrospective; Single-centre; Only TZP-susceptible isolates included; Mostly non-urinary/biliary source; ESBL detected by phenotypic methods only; No screening for AmpC beta lactamases
Benanti 2019 [29]	2008–2015	USA	Retrospective cohort study	103 (21 TZP, 40 cefepime, 42 carbapenem)	ESBL *E. coli*	In patients with haematological malignancy, no differences in 14-day mortality. Prolonged fever and persistent bacteraemia more common with TZP.	Retrospective; Small TZP sample size; Majority of empirical TZP switched to definitive carbapenem after median time of one day
Retamar 2013 [30]	2013	Spain	Retrospective analysis of 6 prospective cohorts	39 (18 low MIC, 21 intermediate/high MIC)	ESBL *E. coli*	For non-urinary source infections treated with TZP, higher 30-day all-cause mortality for intermediate/high vs. low MIC.	Retrospective; Small sample size; No propensity matching; Integration of cohorts with different methodologies
Rodriguez-Baño 2012 [31]	2006–2010	Spain	Retrospective analysis of 6 prospective cohorts	103 empirical therapies (35 TZP, 37 amoxicillin-clavulanic acid, 31 carbapenem)	ESBL *E. coli*	No difference in 30-day mortality.	Retrospective; Underpowered for subgroup analyses; Only TZP-susceptible infections included; Mostly urinary or biliary source; Integration of cohorts with different methodologies
Harris 2018 [32]	2014–2017	Australia, New Zealand, Singapore, Italy, Turkey, Lebanon, South Africa, Saudi Arabia, Canada	RCT	378 (187 TZP, 191 meropenem)	ESBL *E. coli* and *K. pneumoniae*	Definitive TZP associated with increased 30-day mortality relative to definitive meropenem.	Underpowered for subgroup analyses; Evaluation of definitive not empirical therapy; Unblinded; Stepdown therapy to imipenem common after day 5; Low illness severity
Henderson 2021 [33]	2014–2017	As above	Post-hoc analysis of RCT (Harris 2018)	320 (157 TZP, 163 meropenem)	ESBL *E. coli* and *K. pneumoniae*	When TZP non-susceptible strains excluded from analysis, no difference in 30-day mortality. ESBL/narrow spectrum OXA co-expression associated with increased TZP MIC and mortality.	As above; Not all original isolates analysed; Difference in mortality between available and non-available isolates
Heng 2018 [34]	2011–2013	Singapore	Retrospective observational study	123 (73 TZP, 50 carbapenem)	ESBL *E. coli* and *K. pneumoniae*	No difference in 30-day mortality. No associations between virulence factors, type or number of AmpC beta-lactamases and 30-day mortality.	Retrospective; Single-centre; Underpowered for subgroup analysis; Did not evaluate MICs
Ko 2018 [35]	2010–2014	Korea (multicentre)	Retrospective cohort study	232 (183 carbapenem, 41 TZP, 8 other non-carbapenem)	ESBL *E. coli* and *K. pneumoniae*	No difference in 30-day all-cause mortality between TZP and carbapenem.	Retrospective; Only TZP-susceptible infections included
Ng 2016 [36]	2011–2013	Singapore (2 centres)	Retrospective cohort study	151 (94 TZP, 57 carbapenem)	ESBL *E. coli* and *K. pneumoniae*	No difference in 30-day mortality or hospital length of stay. Lower 30-day acquisition of MDR and fungal infections with empirical TZP.	Retrospective; Only carbapenem and TZP-susceptible infections included (but MIC not tested); TZP infusion regimen unspecified; Mostly urinary source
Tsai 2014 [37]	2005–2012	Taiwan (multicentre)	Retrospective cohort study	40 (13 TZP, 21 carbapenem, 6 other antibiotics)	ESBL *Proteus mirabilis*	No difference in 30-day or in-hospital mortality. In the TZP group, 30-day mortality was lower for low vs. high MIC.	Retrospective; Small sample size; No distinction between empirical and definitive TZP; ESBLs identified by phenotypic methods only; No screening for AmpC beta lactamases
Anaerobes							
Ugarte-Torres 2018 [38]	2009–2015	Canada	Retrospective cohort study	95 (36 TZP monotherapy, 49 other antibiotic combinations, 10 no antibiotics)	*Eggerthella lenta*	Empirical TZP monotherapy associated with 30-day mortality and ICU stay. High MIC correlates with higher mortality.	Retrospective; Small sample size; No established TZP MIC breakpoints for *E. lenta*; High proportion of polymicrobial infections
Other							
Bucaneve 2014 [39]	2008–2010	Italy (multicentre)	RCT	180 (86 TZP + tigecycline, 94 TZP monotherapy)	Various: 49% gram-positives (mostly coagulase-negative *Staphylococci*), 35% gram-negatives (mostly *E. coli*), 16% polymicrobial	Higher rates of clinical and microbiological resolution for tigecycline + TZP vs. TZP monotherapy in febrile neutropaenic patients.	Underpowered to detect differences in mortality; unblinded; high rates of MDR organisms

**Table 2 healthcare-10-00851-t002:** Studies reporting TZP toxicity in septic patients.

Reference	Time Period	Setting	Study Design	Sample Size	Reported Toxicity	Key Findings	Limitations
Anand 2011 [40]	2011	USA	Case report	1	Drug-induced thrombocytopaenia	An acutely ill patient with suspected sepsis developed thrombocytopaenia associated with vancomycin and piperacillin-dependent and non-drug-dependent platelet-reactive antibodies following vancomycin/TZP combination therapy. Platelet count recovered only after TZP- cessation.	Case report; Possible multiple interacting aetiologies of thrombocytopaenia
Macwilliam 2012 [41]	2012	UK	Case report	1	Drug-induced thrombocytopaenia	A septic patient developed thrombocytopaenia four days after discontinuation of an 18-day TZP course, and which resolved with IVIg treatment.	Case report; No definitive diagnosis of TZP-induced thrombocytopaenia; Potential exposure to other causative drugs; Unable to reproduce with repeated TZP exposure
Beumier 2015 [42]	2010–2011	Belgium	Retrospective observational	85 TZP trough concentrations	Neurotoxicity	Increasing TZP trough concentration weakly correlated with worsening neurological status.	Retrospective study; Single-centre; Only trough but not peak/steady state drug levels reported; GCS changes used as a surrogate for neurotoxicity
Hall 2019 [43]	2008–2011	USA	Retrospective cohort study	292 (122 TZP, 170 other antibiotics)	Nephrotoxicity	In patients with gram-negative bacteraemia, receiving TZP or treatment duration not associated with nephrotoxicity.	Retrospective study; Single-centre; TZP infusion time and concomitant receipt of vancomycin not documented; Serum creatinine rise used as surrogate for nephrotoxicity

**Table 3 healthcare-10-00851-t003:** Studies reporting on clinical outcomes related to TZP pharmacokinetic and dosing considerations in sepsis. An additional case report [44] can be found in Appendix A.

Reference	Time Period	Setting	Study Design	Sample Size	Key Findings	Limitations
Carrie 2018 [45]	2016–2017	France	Prospective observational study	59	Piperacillin underexposure in patients with augmented renal clearance (creatinine clearance > 170 mL/min) was not significantly associated with therapeutic failure.	Observational study; Single-centre; Significant variability in TZP steady-state concentrations within and between patients; Limited pharmacokinetic model.
Tannous 2020 [46]	2012–2018	Israel	Retrospective observational study	78	In *P. aeruginosa* bacteraemia, spending ≥60% of the dose interval over the MIC is a significant predictor of in-hospital survival.	Retrospective study; Single-centre; Small sample size; TZP concentrations not measured (estimated from a population pharmacokinetic model)
Cutro 2014 [47]	2009–2012	USA	Retrospective cohort study	843 (662 extended infusion, 181 standard infusion)	No significant difference in inpatient mortality and hospital/ICU length of stay. TZP therapy duration significantly shorter in extended infusion group. Mortality rates lower with extended infusion when source presumed to be urinary or intra-abdominal.	Retrospective study with historical matched controls; Single-centre; Did not report on concomitant use of other antimicrobials
Fan 2017 [48]	2013–2015	Hong Kong	RCT	367 patients (224 bacteraemic—108 extended infusion, 116 non-extended infusion)	In bacteraemic patients, 14-day mortality was significantly reduced in the extended infusion group.	Low average renal function may have masked differences in outcome between groups; Unblinded; Underpowered to detect differences in mortality, especially for subgroup analyses
Gonçalves-Pereira 2012 [49]	2006–2010	Portugal (multicentre)	Retrospective cohort study	346 patients (173 matched pairs)	No difference in 28-day mortality, hospital/ICU mortality or length of stay between continuous and intermittent TZP infusion as definitive therapy, even in the most critically ill.	Retrospective study; No data available on TZP concentrations or MIC

## Data Availability

Not applicable.

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
