# Peer review of "Is Piperacillin-Tazobactam an Appropriate Empirical Agent for Hospital-Acquired Sepsis and Community-Acquired Septic Shock of Unknown Origin in Australia?"

_healthcare, 2022, doi:10.3390/healthcare10050851_

Round 1

Reviewer 1 Report

An interesting analysis of the piperacillin/tazobactam suitability as an empirical antibiotic for the treatment of hospital-acquired sepsis and community-acquired septic shock of unknown origin is presented.

I consider the manuscript to be adequately prepared and included studies have been appropriately selected. However, I have a few comments and consider it appropriate to resolve these before final acceptance.

1) I recommend that the authors comment on the question of whether to determine the mechanism of resistance in identified bacterial pathogens (especially members of Enterobacterales), resp. in the case of the production of broad-spectrum beta-lactamases, their type.

2) Do the authors recommend modifying antibiotic treatment based on knowledge of the type of beta-lactamase?

3) Do the authors recommend determining the possible inducible production of AmpC in ESCAPPM?

4) In my opinion, the conclusion of the article is very general. I encourage the authors to consider adding clear recommendations for the use of TZP in the treatment of hospital-acquired sepsis and community-acquired septic shock of unknown origin.

Author Response

In order to address this reviewer’s comments, a search of the literature was performed on 26 April 2022 using key words ‘piperacillin-tazobactam’ and ‘ESBL’ or ‘AmpC’. The methods section has been amended to reflect this. Four additional studies were included in the results and discussion sections of our manuscript, as they met inclusion criteria and allowed us to better address reviewer 2’s suggestions. In addition, our reference list was updated to include the latest version of the Australian Group on Antimicrobial Resistance (AGAR) Sepsis outcome program report. The relevant statistics in the introduction were updated accordingly. Individual comments are addressed below:

Comment 1: I recommend that the authors comment on the question of whether to determine the mechanism of resistance in identified bacterial pathogens (especially members of Enterobacterales), resp. in the case of the production of broad-spectrum beta-lactamases, their type.

Author response to comment 1: To address this comment, we discussed a post-hoc analysis of the MERINO trial (Harris et al. 2018). Their finding of increased mortality for TZP-treated sepsis caused by Enterobacteriaceae with ESBL and narrow spectrum OXA beta lactamase co-expression highlights the potential importance of determining beta lactamase type to guide the selection of appropriate definitive antibiotics. However, we emphasise that the sample size was inadequate to further explore an association between mortality and beta lactamase type.

Comment 2: Do the authors recommend modifying antibiotic treatment based on knowledge of the type of beta-lactamase?

Author response to comment 2: As discussed in our response to comment 1, in the absence of further evidence supporting an association between beta lactamase type and clinical outcomes with TZP therapy, we advise that meropenem should continue to be used first-line for sepsis caused by ESBL producing Enterobacteriaceae regardless of beta-lactamase type.

Comment 3: Do the authors recommend determining the possible inducible production of AmpC in ESCAPPM?

Author response to comment 3: To address this comment, we discussed a post hoc analysis and two additional retrospective cohort studies that explored the clinical outcomes with TZP therapy vs other antibiotics (mostly carbapenems) in sepsis caused by potential AmpC producers. These additional studies emphasised the need for investigation of the clinical outcomes associated with level of AmpC production and AmpC genotype. Therefore, pending further evidence, we do not recommend modifying antibiotic therapy based on the results of AmpC genotyping or AmpC expression levels.

Comment 4: In my opinion, the conclusion of the article is very general. I encourage the authors to consider adding clear recommendations for the use of TZP in the treatment of hospital-acquired sepsis and community-acquired septic shock of unknown origin.

Author response to comment 4: We carefully considered the available literature and concluded that the evidence surrounding our specific research question provides insufficient equipoise to recommend a change to current clinical practice.  To better assess the appropriateness of TZP specifically in the setting of hospital-acquired sepsis and community-acquired septic shock of unknown origin, we stress the importance of conducting a prospective evaluation of TZP appropriateness in the Australian context.

Reviewer 2 Report

Summary of the research and overall impression

The authors Gage-Brown et al. present a well-written review “Is piperacillin-Tazobactam an Appropriate Empirical Agent for Hospital-Acquired Sepsis and Community-Acquired Septic Shock of Unknown Origin in Australia?” of the literature on TZP in hospital-acquired sepsis of unknown origin. Studies addressing this particular research question were sparse and the research question was broadened accordingly. The authors clearly highlight that there is an urgent need for targeted studies to address the research question, as there is a paucity of available studies to address this issue. The need to address the appropriateness of TZP as first-line treatment in response to sepsis (of unknown origin) was clearly highlighted due to the development of novel mechanisms of resistance in gram-negative bacteria. In response to rising levels of antibiotic resistance, addressing this research question is important and therefore, the review contributes to the scientific literature.

The authors also highlight the limitations of the review, which include a lack of separation between the terms hospital versus community-acquired sepsis and sepsis versus septic shock, which is necessary given their lack of distinction in the scientific literature.

Specific areas of improvement

Minor points:

Phrasing:

  • The tables are nicely structured to highlight the effect on different pathogens. However, you often describe in your tables that ZTP is not associated with increased mortality. While important, the more appropriate information to describe would be whether ZTP is effective in treating sepsis in these patients. If possible (providing that data available from original study), I would recommend to describe whether ZTP was associated with decreased mortality (efficacy of the drug) rather than increased mortality (ZTP-associated side-effects) to align it with the main focus of the review.
  • Please ensure that there is correct spacing between the numerical value and the unit symbol. You have used a mix between no space and a space between, e.g., 5mg and 5 mg. Please ensure that a space is inserted throughout.

Author Response

Comment 1: The tables are nicely structured to highlight the effect on different pathogens. However, you often describe in your tables that ZTP is not associated with increased mortality. While important, the more appropriate information to describe would be whether ZTP is effective in treating sepsis in these patients. If possible (providing that data available from original study), I would recommend to describe whether ZTP was associated with decreased mortality (efficacy of the drug) rather than increased mortality (ZTP-associated side-effects) to align it with the main focus of the review.

Author response to comment 1: Where possible, in the ‘key findings’ column of Table 1 we have rephrased 'TZP not associated with increased mortality' to 'no difference in mortality'. However, as there were no instances where TZP was associated with decreased mortality, it would not be factual to rephrase it this way.

Comment 2: Please ensure that there is correct spacing between the numerical value and the unit symbol. You have used a mix between no space and a space between, e.g., 5mg and 5 mg. Please ensure that a space is inserted throughout.

Author response to comment 2: The manuscript has been rectified so that correct spacing has now been placed between the numerical value and the unit symbol.

Round 2

Reviewer 1 Report

I would like to thank the authors for editing the manuscript based on my comments.
I have no further comments and I am happy to recommend the manuscript for acceptance.